# Morbidity and Mortality in Non-Obese Compared to Different Classes of Obesity in Patients Undergoing Transtibial Amputations

**DOI:** 10.3390/jcm12010267

**Published:** 2022-12-29

**Authors:** Senthil Sambandam, Syed Muhammad Mashhood Ali Bokhari, Shirling Tsai, Vishaal Sakthivel Nathan, Tejas Senthil, Heather Lanier, Sergio Huerta

**Affiliations:** 1Department of Orthopedics, VA North Texas Health Care System, Dallas, TX 75216, USA; 2Department of General Surgery, VA North Texas Health Care System, Dallas, TX 75216, USA; 3Department of Vascular Surgery, VA North Texas Health Care System, Dallas, TX 75216, USA

**Keywords:** transtibial amputation, BKA, transfemoral amputation, AKA, National Inpatient Sample

## Abstract

This study assesses the effect of obesity classes on outcomes and inpatient-hospital-cost compared to non-obese individuals undergoing below-knee amputations (BKAs). Retrospective matched-case controlled study performed on data from NIS Database. We identified three groups: N-Ob (BMI < 29.9 kg/m^2^; *n* = 3104), Ob-I/II (BMI: 30 to 39.9 kg/m^2^; *n* = 3107), and Ob-III (BMI > 40; *n* = 3092); matched for gender, comorbidities, tobacco use and elective vs. emergent surgery. Differences in morbidity, mortality, hospital length of stay (LOS), and total inpatient cost were analyzed. Blood loss anemia was more common in Ob-III compared to Ob-I/II patients (OR = 1.2; 95% CI = 1.1–1.4); blood transfusions were less commonly required in Ob-I/II (OR = 0.8; 95% CI = 0.7–0.9) comparatively; Ob-I/II encountered pneumonia less frequently (OR = 0.9; 95% CI = 0.4–0.9), whereas myocardial infarction was more frequent (OR = 7.0; 95% CI = 2.1–23.6) compared to N-Ob patients. Acute renal failure is more frequent in Ob-I/II (OR = 1.2; 95% CI = 1.0–1.3) and Ob-III (OR = 1.8; 95% CI = 1.6–1.9) compared to the N-Ob cohort. LOS was higher in N-Ob (13.1 ± 12.8 days) and Ob-III (13.5 ± 12.4 d) compared to Ob-I/II cohort (11.8 ± 10.1 d; *p* < 0.001). Mortality was 2.8%, 1.4%, and 2.9% (*p* < 0.001) for N-Ob, Ob-I/II, and Ob-III, respectively. Hospital charges were $22,025 higher in the Ob-III cohort. Ob-I/II is protective against peri-operative complications and death, whereas hospital cost is substantially higher in Ob-III patients undergoing BKAs.

## 1. Introduction

The morbidity and mortality of major lower extremity amputations (mLEAs) have been high for the past several decades and have not considerably changed in the 21st century [1].

Major lower extremity amputations include amputations through the femur (AKA) and tibia (BKA). Morbidity and mortality are uniformly higher for patients submitting to an AKA compared to a BKA (1). For patients with class I/II obesity (Ob-I/II; BMI 30–39.9 kg/m^2^) and patients with class III obesity (Ob-III; BMI > 40.0 kg/m^2^) undergoing mLEA, it is important to determine the potential rate of complications and mortality. Studies comparing outcomes in non-obese (N-Ob), Ob-I/II, and Ob-III patients undergoing mLEA are limited. A major challenge in comparing outcomes of mLEAs between these groups emanates from a wide range of differences in demographics and pre-existing comorbidities that might affect outcomes and mortality differently [2,3]. For instance, patients with obesity are more likely to suffer from diabetes compared to non-obese individuals, which might lead to a higher rate of surgical site infections, wound dehiscence, and blood loss [3]. On the other hand, non-obese patients are more likely older and tobacco users [4], which might lead to a higher rate of cardio-pulmonary complications [5]. Post-operative outcomes will differ depending on the underlying conditions of a cohort of patients presenting for an operation.

The “obesity paradox” is a phenomenon where a moderate increase in BMI (overweight and class I obesity) is protective against adverse outcomes in patients with heart disease undergoing cardiac interventions [6,7,8]. However, this trend has not been described for various classes of obesity in a patient undergoing BKAs.

In this study, we examined a large national database and matched N-Ob, Ob-I/II, and Ob-III patients by multiple variables undergoing BKAs. The aim of this study was to assess the risk of Ob-I/II and Ob-III in patients undergoing BKAs. We hypothesized that Ob-III would lead to worse outcomes in patients undergoing BKAs compared to N-Ob and Ob-I/II subjects undergoing the same operative procedure.

## 2. Materials and Methods

The present analysis is a retrospective, matched case-control study that used the National Inpatient Sample (NIS) Database from 2016 to 2019. The NIS database includes data from more than 7 million hospital stays and encompasses 20% of the hospitals in the United States. NIS data are verified using a quality assessment evaluation comparing data points to standardized normative values by an independent contractor.

Data elements within the NIS Database include patient demographic characteristics, comorbidities, inpatient complications, LOS, source of payment, hospital charges, and discharge status. The database 2016 to 2019 version utilizes the International Classification of Diseases, Tenth Revision, Clinical Modification/Procedure Coding System (ICD-10-CM/PCS) [9].

We divided the patient into three groups based on their body mass index (BMI), which is a marker of the percentage of body fat mass [10]: N-Ob (BMI < 29.9 kg/m^2^), Ob-I/II (BMI 30–39.9 kg/m^2^), and Ob-III [BMI > 40 kg/m^2^). Patients with class I and II obesity were grouped together to match the three cohorts by size and to assess the role of class III obesity compared to Ob-I/II patients and the N-Ob cohort.

### 2.1. Data Acquisition

This study was exempt from IRB approval because the data are de-identified and publicly available. Patients who underwent BKAs were identified using the ICD-10 procedural codes. We identified 31,236 patients who had amputations. From this cohort, we included 6199 patients who underwent BKA, and their complete data set was available through the database. The remaining patients were excluded if they did not undergo BKA, were undergoing revision of a previous amputation, had previously undergone a foot amputation, or the available data were incomplete. Patients with obesity were further classified into Ob-I/II (*n* = 3107) and Ob-III (*n* = 3092). Ob-I/II, Ob-III, and N-Ob (*n* = 3104) were matched with respect to age, sex, diabetes, tobacco use, and emergency versus elective surgery. After matching, post-operative medical and surgical outcomes were aggregated from the NIS database using ICD 10 codes. The outcomes included mortality, systemic complications such as postoperative anemia, acute renal insufficiency (AKI), pulmonary embolism (PE), deep vein thrombosis (DVT), myocardial infarction (MI), pneumonia (PNA), cardiac arrest, and blood transfusion. Local complications were also cataloged and included wound dehiscence, superficial surgical site infection (SSI), and deep SSI. Additionally, hospital LOS and average inpatient cost per patient of stay were included in the analysis.

### 2.2. Statistical Analysis

All statistical analyses were conducted using SPSS version 27.0 (IBM; Armonk, NY, USA). Originally, descriptive statistics were used to aggregate patient demographic data. Propensity score matching was done using SPSS software, and matching performance was checked by performing a chi-square test on matched variables. N-Ob and Ob-III were compared to Ob-I/II patients’ *t*-tests when analyzing numerical variables. Chi-squared analyses were used when analyzing binomial variables. A *p*-value < 0.05 was considered significant for all tests. Odds ratios (OR) and their corresponding 95% confidence intervals (95% CI) for the surgical outcomes and complications were measured as a ratio of the incidence in the case group to the incidence in the control group.

## 3. Results

### 3.1. Patient Demographics

Patient demographics for patients undergoing BKAs in each group are depicted in Table 1. Matching was performed for age, gender, diabetes, tobacco use, and type of surgery (elective vs. emergent).

### 3.2. Morbidity and Mortality in Patients Undergoing BKA

Ob-III patients were more likely to have blood loss anemia compared to Ob-I/II patients (OR = 1.24; 95% CI = 1.11 to 1.40). Ob-I/II patients had a lower risk for requiring inpatient blood transfusions than N-Ob and Ob-III patients (OR = 0.83; 95% CI = 0.73 to 0.94). Pneumonia occurred less frequently in Ob-I/II patients compared to N-Ob patients (OR = 0.85; 95% CI = 0.39 to 0.96). Myocardial infarction (OR = 7.0; 95% CI = 2.1 to 23.6) was substantially more common in Ob-I/II patients compared to the N-Ob cohort. Compared to the N-Ob cohort, AKI occurred more commonly in Ob-I/II (OR = 1.2; 95% CI = 1.0 to 1.3) and Ob-III patients (OR = 1.8; CI = 1.6 to 1.9). Hospital LOS was over one day higher in N-Ob and Ob-III patients compared to Ob-I/II patients. Similarly, inpatient mortality was higher for both N-Ob patients and Ob-III (~3.0%) compared to Ob-I/II patients (1.4%). While total inpatient hospital charges per patient were similar for N-Ob and Ob-I/II patients, they were $22,025 higher for Ob-III patients. These results are summarized in Table 2.

## 4. Discussion

The global increase in obesity is associated with diabetes mellitus, hypertension [11], and cardiovascular disease [12,13], as well as all-cause mortality [14]. However, the “obesity paradox” describes a condition where obesity is protective against some adverse outcomes. For instance, obesity was protective against adverse outcomes in patients with heart failure [6,15]. Patients undergoing coronary artery bypass surgery [7] and patients are suffering from myocardial infarctions [8].

For patients undergoing vascular surgery, the “obesity paradox” has produced mixed observations. For instance, in a small retrospective study of patients undergoing major reconstructive vascular surgery, overweight and patients with obesity had a higher rate of surgical site infections [16]. Another study analyzed prosthetic use, ambulation, functional independence, and survival in obese vs. non-obese patients with peripheral vascular disease submitting to mLEAs [17,18]. There was no difference in any of these factors between obese and non-obese patients. Thus, one study showed worse outcomes in patients with obesity, and the other one showed no difference. There was no “obesity paradox” in these studies.

However, in 7543 patients undergoing aneurysm repair, cerebrovascular procedures, and amputations, morbidity and mortality demonstrated a U-shaped or a J-shaped distribution depending on BMI status [2]. Overweight and patients with obesity had a protective effect for complications, whereas the highest risk for adverse outcomes was observed in both extremes: underweight and patients with class III obesity. Similarly, another study in patients undergoing major lower extremity endovascular interventions found obesity to be protective for 30-day complication rates [19].

In the current study, we were less interested in underweight patients, and the goal was to assess the role of class III obesity compared to obese and N-Ob individuals. We extracted data from the NIS database (the largest inpatient database in the United States [9]) from 2016 to 2019 in all patients undergoing BKAs. We corrected several variables (emergency status, gender, age, diabetes, and smoking status) such that the effect of class I/II obesity and class III obesity in a mid- to high-risk operation could be ascertained along with its associated cost. To our knowledge, this is the first study assessing the role of class I/II obesity and class III obesity in patients undergoing BKAs.

Our analysis produced results consistent with some studies showing obesity being protective against adverse outcomes following vascular interventions [2,19]. The present study demonstrated a U-shaped distribution for mortality, with Ob-I/II being protective (1.4%) compared to N-Ob (2.8%) and Ob-III patients (2.9%). Hospital LOS demonstrated a similar pattern: Ob-I/II patients’ hospital LOS was a day less compared to both the N-Ob and the Ob-III cohorts. The relationship between other outcomes between N-Ob, Ob-I/II, and Ob-III patients undergoing BKA showed mixed results. Outcomes such as peri-operative blood transfusion and development of PNA were less frequent in the Ob-I/II cohort, demonstrating a protective effect, whereas complications such as MI and AKI were more commonly seen in the Ob-I/II cohort. Despite the difference in outcomes, overall data suggested that inpatient hospital cost for N-Ob and Ob-I/II was similar. Hospital charges were $22,024.9 higher in the Ob-III cohort compared to the other two, which were likely driven by blood loss and renal insufficiency leading to a longer LOS. Superficial and deep SSI were low and similar in all cohorts undergoing BKA.

The relationship in outcomes between these groups is complex and emanates from the wide variety of pre-existing confounding variables in each cohort that differs between them and predisposes them to morbidity and mortality differently [3]. The “obesity paradox” leading to the protective effects of mild obesity in a wide range of vascular operations has been previously explored in detail by Davenport et al. [2]. In their manuscript, the authors outlined that the paradox in outcomes and obesity might be related to the metabolic activity of adipose tissue and the array of associated hormones that might protect against worse outcomes. They also noted that serum albumin levels differed widely in different BMI groups [2]. In our study, the “obesity paradox” applies to perioperative mortality and adverse events following surgery for Ob-I/II group. However, the Ob-III group was noted to have worse outcomes in many of the variables measured: worse mortality, higher hospital LOS, and higher in-hospital cost compared to the other cohorts. The “obesity paradox” does not apply to Ob-III patients undergoing BKAs.

The following study has several limitations. This study only assessed in-patient outcomes. Although obesity might be protective for some outcomes in the short term, the long-term effects in patients with obesity undergoing BKAs are unclear. Furthermore, it has been noted that once an amputation is performed, amputees become more obese compared to the general population [20], and the effects of this have not been examined in the long term. While we matched several important variables in the study, the NIS database does not contain granular data such as serum albumin levels to match other important variables and assess the true contribution of obesity to outcomes in patients undergoing BKAs. A major limitation of our study is also that the NIS database cannot determine when a secondary event occurred except when it is acute. This is an observational study, and while there might be an association between the factors that we examined and an increase in morbidity, hospital LOS, and hospital cost, a causal link cannot be established.

Overall, this study establishes the basic framework for managing patients with obesity undergoing BKA in the peri-operative setting. We recommend further randomized controlled trials evaluating the factors we have identified in the study to establish causality.

## 5. Conclusions

After controlling for several variables, Ob-I/II is protective against a limited number of complications and risk of peri-operative death in patients undergoing BKA. Hospital LOS and inpatient costs were higher in Ob-III patients compared with Ob-I/II and N-Ob patients, likely because of the higher frequency of blood loss anemia and subsequent blood transfusions, as well as AKI.

## Figures and Tables

**Table 1 jcm-12-00267-t001:** Patient demographics in patients undergoing below-the-knee amputations.

	N-Ob ^1^ (*n* = 3104)	Ob-I/II ^2^ (*n* = 3107)	Ob-III ^3^ (*n* = 3092)
Age [y.o. ± (SD)]	59.0 (13.3)	59.4 (12.0)	59.1 (11.8)
Age category	Matched	Matched	Matched
Female [*n* (%)]	971 (31.3)	972 (31.3)	965 (31.2)
Diabetes [*n* (%)]	56 (1.8)	56 (1.8)	53 (1.7)
Tobacco use [*n* (%)]	104 (3.4)	104 (3.4)	102 (3.3)
Elective versus Emergent [*n* (%)]	840 (27.1)	840 (27.0)	833 (26.9)

^1^ N-Ob = Non-Obese (BMI < 29.9 kg/m^2^); ^2^ Ob-I/II = Class I and II obesity (BMI 30–39.9 kg/m^2^), and ^3^ Ob-III = Class III obesity (BMI > 40 kg/m^2^).

**Table 2 jcm-12-00267-t002:** Morbidity and mortality of N-Ob, Ob-I/II, and Ob-III patients undergoing below-the-knee amputations. The average total cost per patient per hospitalization is also included. The odds ratio has been adjusted for age, sex, diabetes, tobacco use, and emergency versus elective surgery.

	N-Ob *(n* = 3104)	Ob-I/II (*n* = 3107)		Ob-III (*n* = 3092)	
	*n* (%)	*n* (%)	Odds Ratio (95% CI)	*n* (%)	Odds Ratio (95% CI)
Blood loss anemia	771 (24.8)	699 (22.5)	0.88 (0.78 to 0.99)	822 (26.6)	**1.24** (1.11 to 1.40)
Blood Transfusion	601 (19.4)	516 (16.6)	**0.83** (0.73 to 0.94)	596 (19.3)	**1.19** (1.05 to 1.36)
Wound Dehiscence	58 (1.9)	59 (1.9)	1.02 (0.71 to 0.94)	71 (2.3)	1.21 (0.85 to 1.72)
Superficial SSI ^1^	0 (0.0)	** (0.13)	NA	** (0.1)	0.50 (0.09 to 2.74)
Deep SSI	0 (0.0)	** (0.19)	NA	** (0.1)	0.33 (0.06 to 1.65)
DVT ^2^	49 (1.6)	49 (1.6)	1.00 (0.67 to 1.45)	55 (1.8)	1.13 (0.76 to 1.66)
PE ^3^	17 (0.6)	13 (0.4)	0.76 (0.37 to 1.57)	22 (0.7)	1.70 (0.85 to 3.39)
PNA ^4^	142 (4.6)	76 (2.5)	**0.85** (0.39 to 0.69)	85 (2.7)	1.12 (0.82 to 1.54)
MI ^5^	** (0.1)	21 (0.7)	**7.03** (2.10 to 23.60)	33 (1.1)	1.58 (0.91 to 2.74)
AKI ^6^	804 (25.9)	901 (29.0)	**1.17** (1.04 to 1.31)	1289 (41.7)	**1.75** (1.57 to 1.94)
LOS ^7^ (Days ± SD ^8^)	13.1 ± 12.8	11.8 ± 10.1	<0.001	13.5 ± 12.4	<0.001
Mortality	87 (2.8)	44 (1.4)	**0.50** (0.35 to 0.72)	90 (2.9)	**2.08** (1.45 to 3.00)
Charges (USD ^9^)	129,271.4	125,880.6	0.443	147,905.6	<0.001

^1^ SSI = surgical site infection, ^2^ DVT = deep venous thrombosis, ^3^ PE = pulmonary embolism ^4^ PNA = pneumonia, ^5^ MI = myocardial infarction, ^6^ AKI = acute renal failure, ^7^ LOS = hospital length of stay, ^8^ SD= standard deviation, ^9^ USD = US dollars. Bold font *p* ≤ 0.05. ** Number between 1 to 10 not reported as per Health Care Utility Project (HCUP) data use agreement.

## Data Availability

The data presented in this study are openly available in the National Inpatient Sample (NIS) Database from 2016 to 2019.

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
