# Peer review of "Morbidity and Mortality in Non-Obese Compared to Different Classes of Obesity in Patients Undergoing Transtibial Amputations"

_jcm, 2022, doi:10.3390/jcm12010267_

Round 1
Reviewer 1 Report
The analysis was a retrospective matched case-control study that used the National Inpatient Sample (NIS) Database of the USA to assess the effect of obesity classes on outcomes and inpatient-hospital-cost compared to non-obese individuals undergoing below-knee amputations. The results of the analysis are similar to the clinical results. We consider two points about this study:
1. The procedure of below-knee amputation is a salvage procedure for a patient with critical or devastating conditions. Are these medical conditions related to obesity?
2. In ordinally, the rate of superficial and deep surgical site infection of surgical procedures in obese is higher than in non-obese individuals. But the higher infection rate in obese-I/II patients accepted the procedure than in obsess-III patients in this study. Can the authors explain the possibility of the results?
Author Response
Reviewer's comment 1: The procedure of below-knee amputation is a salvage procedure for a patient with critical or devastating conditions. Are these medical conditions related to obesity?
Response: For our study we did not assume that these conditions are related to obesity, therefore, we compared the data between patients with normal BMI vs obesity classes to assess the effect of BMI on rate of complications.
Reviewer's comment 2: In ordinally, the rate of superficial and deep surgical site infection of surgical procedures in obese is higher than in non-obese individuals. But the higher infection rate in obese-I/II patients accepted the procedure than in obsess-III patients in this study. Can the authors explain the possibility of the results?
Response: Our p-value was not significant for deep and superficial SSI while comparing the three groups. We mentioned in the discussion, “Superficial and deep SSI were low and similar in all cohorts undergoing BKA.”
Reviewer 2 Report
This is a very interesting and important study to determine mortality and in-hospital complications differences between non-obese patients/ obesity class I/II patients and obesity type III patients.
My main question/concern for the paper is: How did the authors determine the timing for any of the complications? With the exception of possibly acute renal failure, the other diagnoses could have happened at any point during the hospitalization or on a past hospitalization when selected as secondary diagnoses. I refer the authors to page 9, Table 1 for the following PDF link to HCUP/NIS: https://www.hcup-us.ahrq.gov/reports/methods/2016-06.pdf. Because NIS does not have a POA variable (compared to state databases), the ability to determine if complications occurred after surgery during the index hospitalization is not possible. Even if there was “post procedural diagnosis” ICD 10 code, it may not be possible to determine which hospitalization the complication occurred and the complication was related to an alternate/similar surgery. Only procedure ICD-PCS codes can reliably be captured for a hospitalization using the NIS (page 10-Table 3 https://www.hcup-us.ahrq.gov/reports/methods/2016-06.pdf.)
Also, given the significant racial disparities for patients with obesity in the United States (https://www.cdc.gov/obesity/data/adult.html) , I am concerned that authors did not match for patient race, especially since Black and Hispanic patients may make up a good portion of patients studied in the NIS. The results may look very different after race has been taken into account.
Very very interesting study. However, the authors should revise/address the above concerns prior to publication.
Author Response
Response 1:
We do acknowledge that NIS data has the limitation of not being able to identify if a particular complication was preexisting or related to this procedure. This has also been listed by several NIS studies and editorial comments as one of the major drawbacks of the NIS database. Still, it continues to be the most widely used administrative database in surgical research and surgeons often use the finding of the NIS studies considering other available clinical evidence. Also, the complications selected by NIS research studies are acute and unlikely to be pre-existing comorbidities since the presence of such acute medical complications (MI, AKI, PE) would often preclude major surgery in most patients.
Response 2:
We thank the reviewers for this comment. Race and racial disparities are exceedingly important in many of our research protocols as well as in patients undergoing major lower extremity amputations. In our previous study inclusive of over 1000 mLEAs we noted the following (1).
Race has been previously found to be associated with mortality following LEAs, affecting black patients negatively compared to Hispanic and non-Hispanic whites (2). A subsequent study found that major vascular operations such as AAA, lower extremity revascularization, and major LEAs were associated with mortality in black patients. However, the multivariable analysis did not identify race as an independent predictor of mortality (3). Other studies including VA patients have found black race to be protective against mortality (4). In a separate study, the non-white race was associated with morbidity, but not mortality (5). In our study, while black race was associated with 5-year mortality, it was not an independent predictor of mortality (1).
In the present paper, we wanted to focus on the role of obesity exclusively. We also wanted to determine how it affected the cost of the hospital healthcare system. Race pertaining to cost in the NIS would not have affected the primary outcome we were seeking. Nonetheless, this will remain an important issue for further studies in our analyses.
- Tsai S LH, Tran N, Pham T, Huerta S. Current Predictors of Mortality in Veteran Patients Undergoing Major Lower Extremity Amputations: Risk Factors Have Not Changed and Mortality Remains High. The American Surgeon. 2022;0(0). doi:10.1177/00031348221074235.
- Lavery LA, Van Houtum, W. H., Armstrong, D. G., Harkless, L. B., Ashry, H. R., & Walker, S. C. (1997). Mortality following lower extremity amputation in minorities with diabetes mellitus. Diabetes research and clinical practice, 37(1), 41-47.
- Collins TC, Johnson M, Daley J, Henderson WG, Khuri SF, Gordon HS. Preoperative risk factors for 30-day mortality after elective surgery for vascular disease in Department of Veterans Affairs hospitals: is race important? J Vasc Surg. 2001;34(4):634-40.
- Feinglass J, Pearce, W. H., Martin, G. J., Gibbs, J., Cowper, D., Sorensen, M., ... & Khuri, S. (2001). Postoperative and late survival outcomes after major amputation: findings from the Department of Veterans Affairs National Surgical Quality Improvement Program. Surgery, 130(1), 21-29.
- Toursarkissian B, Shireman, P. K., Harrison, A., & D'Ayala, M. (2002). Major lower-extremity amputation: contemporary experience in a single Veterans Affairs institution. The American Surgeon, 68(7), 606.
Round 2
Reviewer 2 Report
Thank you again for the opportunity to review the paper. As pointed out in the previous review, NIS cannot determine when a secondary event occurred (with an exception possibly being acute)- I refer the authors again to the following pdf https://www.hcup-us.ahrq.gov/reports/methods/2016-06.pdf. Page 9 Table 1 where "sepsis" is being used as the example. The NIS does not have a POA variable. Therefore, myocardial infarction (similar to the sepsis example used by the HCUP) can not be determined on when the event occurred. The POA variable is only in the state database.
Author Response
We thank the reviewer for the thoughtful comments on our manuscript.
We agree with your comment and have included in the manuscript the following statement in the limitations of the study: “A major limitation of our study is also that the NIS database cannot determine when a secondary event occurred except when it is acute.”
However, we believe that the same limitation applies to all the cohorts of non-obese, obese, and MO. Yet, this remains a limitation of our study that requires further studies with databases that contain more granular data.
We derived the idea of collecting these variables from the NIS database based on literature review of studies that have utilized a similar technique such as a study by Kandregula et al, 2021 (1) and Anastasio et al, 2021 (2).
- Kandregula S, Birk HS, Savardekar A, Newman WC, Beyl R, Trosclair K, et al. Spinal Fractures in Ankylosing Spondylitis: Patterns, Management, and Complications in the United States - Analysis of Latest Nationwide Inpatient Sample Data. Neurospine. 2021;18(4):786-97.
- Anastasio AT, Patel, P. S., Farley, K. X., Kadakia, R., & Adams, S. B. (2021). Total ankle arthroplasty and ankle arthrodesis in rheumatic disease patients: An analysis of outcomes and complications using the National Inpatient Sample (NIS) database. Foot and Ankle Surgery, 27(3), 321-325.